# Operando magnetic resonance imaging for mapping of temperature and redox species in thermo-electrochemical cells

Isuru E. Gunathilaka [1], Jennifer M. Pringle [2] & Luke A. O'Dell [1✉]

Low-grade waste heat is an abundant and underutilised energy source. In this context, thermo-electrochemical cells (i.e., systems able to harvest heat to generate electricity) are being intensively studied to deliver the promises of efficient and cost-effective energy harvesting and electricity generation. However, despite the advances in performance disclosed in recent years, understanding the internal processes occurring within these devices is challenging. In order to shed light on these mechanisms, here we report an operando magnetic resonance imaging approach that can provide quantitative spatial maps of the electrolyte temperature and redox ion concentrations in functioning thermo-electrochemical cells. Time-resolved images are obtained from liquid and gel electrolytes, allowing the observation of the effects of redox reactions and competing mass transfer processes such as thermophoresis and diffusion. We also correlate the physicochemical properties of the system with the device performance via simultaneous electrochemical measurements.

[1] ARC Centre of Excellence for Electromaterials Science (ACES), Institute for Frontier Materials, Deakin University, Geelong Waurn Ponds Campus, Victoria 3220, Australia. [2] ARC Centre of Excellence for Electromaterials Science (ACES), Institute for Frontier Materials, Deakin University, Melbourne Burwood Campus, Victoria 3125, Australia. ✉email: luke.odell@deakin.edu.au

As the demand for technology-based solutions to climate change and improved renewable energy sources continues to accelerate, the utilisation of low-grade waste heat becomes increasingly attractive. Such heat is generated by industrial processes, automobiles, natural sources (e.g., solar or geothermal) and even the human body, and represents a vast and largely untapped energy resource[1]. Thermo-electrochemical cells (also referred to as thermogalvanic cells, and hereafter as thermocells) are devices capable of generating electricity from waste heat[2,3]. In their simplest form, they consist of two parallel plate electrodes and an electrolyte containing a redox couple with a temperature-dependent redox potential. Under an applied thermal gradient, either oxidation or reduction will dominate at the hot electrode depending on the sign of the reaction entropy[4], with the opposite process occurring at the cold electrode. This generates an electric potential across the cell (the Seebeck effect), and when an external load resistance is connected to the electrodes the drawn current will drive further redox reactions, resulting in the continuous generation of power with no fuel consumption or emissions for as long as the temperature gradient is maintained. Thermocells have many advantageous features such as no moving parts or potentially deleterious effects such as plating, stripping, interphase formation or intercalation processes, which will be beneficial for longevity, and they utilise much lower cost materials than traditional semi-conductor based thermoelectrics, making them promising for large scale, low-cost applications. Recently, significant progress has been made in developing wearable thermocells to harvest body heat[5–7], as well as in developing new thermocell designs that incorporate separators or membranes[8], material phase transitions[9,10], crystallisation processes[11], and improved electrode[12] and electrolyte[13–15] materials to boost their Seebeck coefficient and efficiency.

Despite the simplicity of their design, there are a large number of distinct physical processes that can occur within the thermocell electrolyte. These include the redox reactions and accompanying changes in ion clustering and solvation, the establishment of local concentration gradients and resulting diffusion along these gradients, self-diffusion (due to Brownian motion), thermophoresis (the migration of species along a temperature gradient, usually from hot to cold, also known as the Soret effect), electrophoresis (the migration of charged species along a potential) and convection. All of these processes will be highly inter-dependent, for example convection can be caused by the applied temperature gradient but also by variations in the electrolyte density resulting from the redox reactions and concentration gradients. Moreover, these processes will also depend on various external parameters such as the shape, size and orientation of the electrolyte chamber, and due to the effects of the chamber walls they will potentially show distinct spatial variations in all three dimensions. This multitude of interacting factors is complex enough during the steady-state operation of the device, but they will also vary as a function of the applied thermal gradient and load resistance, both of which may be time dependent in real-world applications. Understanding and quantifying these phenomena, their relation to the cell design and their impact on the performance of the device is therefore a major challenge.

Thus far, the most viable way to visualise these processes has been via numerical modelling, and a number of previous publications have used this approach to predict the spatial variations in temperature and concentrations of the redox species for cells operating under steady-state conditions[11,16–19]. These studies have provided valuable insights, for example Sokirko's model predicted linear concentration gradients for the redox species in the absence of convection[16], and, when convection is present, the existence of diffusive layers at the electrodes (regions of enhanced concentration of redox species due to their generation at the electrode surface) with a neighbouring "inversion layer" showing a reduced concentration of that species relative to the bulk[16,17]. These features are predicted to be more pronounced at the cold electrode due to the slower local diffusion. Salazar and co-workers used their model as a basis for the optimisation of cell designs and geometries, and showed that stacking cells in series could significantly boost conversion efficiency[17]. However, such models inevitably necessitate various simplifications. For example, Salazar's model was restricted to two dimensions, thus could not consider any effects of the interior side walls of the cell, and also had to neglect the effects of thermophoresis under the assumption that other mass transfer effects are dominant in a functioning cell[17].

To the best of our knowledge, the experimental observation of the various inter-related and spatially-dependent phenomena occurring in a working thermocell has not yet been achieved. The vast majority of experimental studies have focused on electrochemical measurements[20] carried out on thermocells featuring different electrode and electrolyte materials, and with different cell geometries, orientations and designs[21–26]. While these are extremely useful and can be used to quantify the device performance, they provide little information on the processes occurring within the electrolyte and certainly no insights into spatial variations in the cell. Said and co-workers have used infrared (IR) thermal imaging to study the temperature distribution across thermocells incorporating polymer membranes[27], while Yu and co-workers also used this method to image temperature distributions in their thermosensitive-crystallisation boosted devices[11]. While IR imaging certainly provides useful data, it only strictly reports on the temperature across the outer surface of the device, while spatial variations in temperature within the electrolyte interior may be hidden.

Operando magnetic resonance imaging (MRI) is a versatile and non-invasive technique that has been used by a number of research groups to study energy storage devices such as double-layer capacitors[28] and batteries[29,30], allowing the mapping of ion concentration gradients in electrolytes[31–34], studies of device charge state[35] and current distributions[36], and the observation of processes such as Li and Na ion intercalation into electrodes[37], the growth of metallic dendrites from the electrode surface[38], and magnetohydrodynamics in the electrolyte[39]. Readers wishing to learn more about operando MRI methods for electrochemical devices are referred to two informative review articles[29,30]. This technique is particularly well-suited for studies of thermocells given the inherent paramagnetic nature of many redox species that will result in enhanced relaxation-based image contrast[40], as well as its ability to quantitatively probe translational molecular dynamics including diffusion and flow.

To this end, we have constructed custom thermocells designed specifically for operando MRI characterisation using a micro-imaging probe with a 25 mm inner diameter radiofrequency coil. The basic cell layouts are shown schematically in Fig. 1, while additional 3D design images are shown in Supplementary Figs. 1–4. The cell housing is made of polyether ether ketone (PEEK), an inert material that is easily machined and is thermomechanically stable. The electrolyte is contained in a central chamber between two 0.25-mm-thick platinum plate electrodes held in place using butyl rubber gaskets. The electrode temperatures are controlled in situ using hot and cold gas streams that flow over the outer surfaces of the electrodes (shown for the vertical electrode cell in Fig. 1a) and whose temperatures and pressures can be individually controlled. Due to the design of the imaging probe used and space restrictions inside the magnet, the hot gas is required to enter the cell from below with the other gas connections located at the top (Supplementary Fig. 5). Despite this restriction, the modular design of the horizontal electrode cell

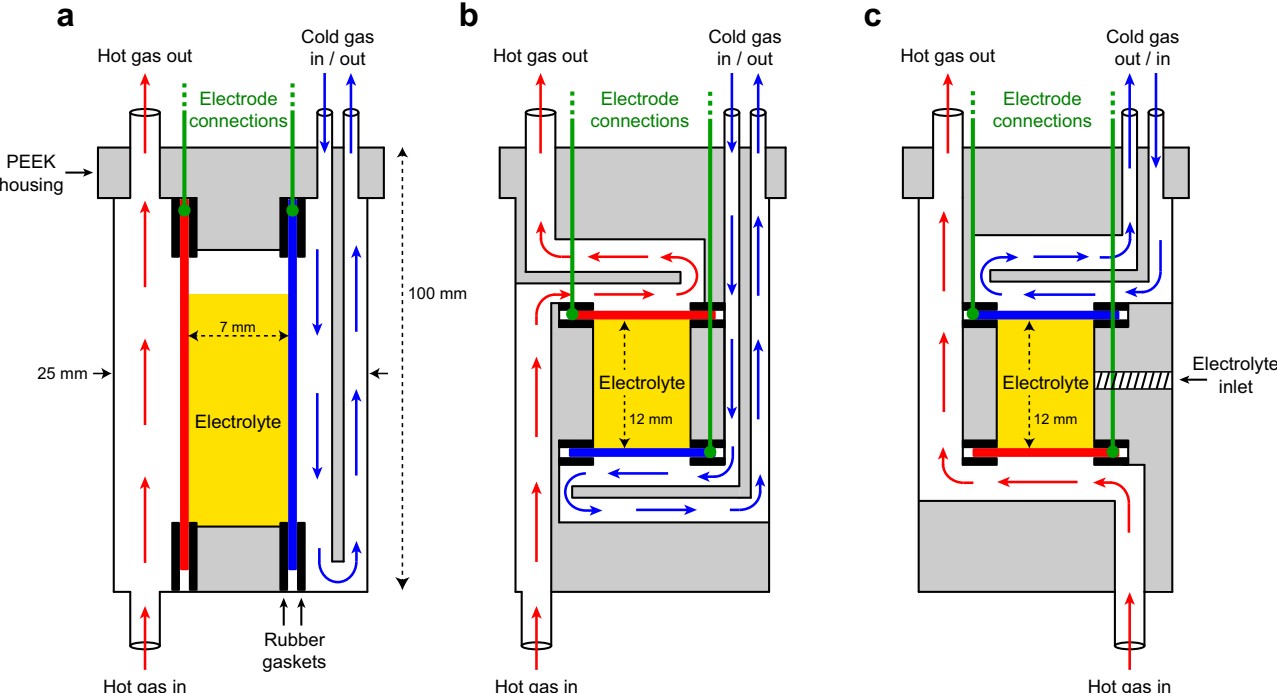

**Fig. 1 Schematics of the thermocell designs for operando MRI experiments. a** Vertical, **b** horizontal hot-above-cold and **c** cold-above-hot electrode orientations. Red and blue arrows represent hot and cold gas flow pathways, respectively. Additional illustrations of the cells are provided in the Supplementary Information, and full design details are available upon request.

allows either hot-above-cold or cold-above-hot orientations (Fig. 1b and c). As the electrolyte temperature is measured directly during the experiments via the MRI images obtained, no thermocouples are required. However, benchtop temperature measurements were also carried out using thermocouples at various positions within the cell (inside the electrolyte, and near the outer surfaces of the electrodes) to verify the reproducibility of the electrolyte temperature as determined by MRI (±2.5 °C) and the stability of the gas flow temperatures over time (±0.5 °C). Electrode connections are achieved using wires extending from the top of the cells and these are connected to a potentiostat located outside of the magnet using shielded cables.

The electrolyte system chosen for this work consists of the tris(bipyridyl)cobalt redox couple $Co(bpy)_3$ with the bis(tri-fluoromethanesulfonyl)-imide (TFSI) anion in the solvent 3-methoxypropionitrile (MPN) (molecular structures shown in Fig. 2a, b and c). The cobalt cation can exist in either the para-magnetic $Co^{2+}$ or diamagnetic $Co^{3+}$ oxidation states, with two or three associated TFSI anions, respectively. This system was cho-sen based on its relatively high Seebeck coefficient of 1.99 mV K$^{-1}$ (measured for an MPN electrolyte containing 0.05 M of both redox species)[23]. A gelled form synthesised by incorporating 5 wt % polyvinylidene difluoride (PVDF) has also been studied previously[25] and was investigated in this work alongside the liquid form. For this electrolyte, the redox reaction entropy is positive and thus the hot electrode acts as the cathode and reduction reactions dominate on that side of the thermocell, forming the paramagnetic $Co^{2+}$ state, with the opposite process occurring at the cold electrode.

## Results

**$^1$H and $^{19}$F image contrast.** Axial-slice images of liquid elec-trolyte samples with different concentrations of each cobalt oxi-dation state, acquired using the $^1$H nuclei of the MPN solvent molecules and the $^{19}$F nuclei of the TFSI anions, are shown in

Fig. 2d and e, respectively. The protons of the bipyridyl ligands give well-resolved $^1$H nuclear magnetic resonance (NMR) signals for the different cobalt oxidation states due to the large para-magnetic shift induced by the $Co^{2+}$ (Supplementary Fig. 6), so chemical shift selective imaging could in principle be used to directly observe these species. However, the MPN solvent mole-cules give $^1$H signals that are approximately two orders of mag-nitude larger and so these signals were exploited for the $^1$H imaging. Figure 2d shows that the MPN $^1$H $T_1$ (i.e., the long-itudinal nuclear spin relaxation time, a parameter commonly used to generate MRI image contrast) is sensitive to the concentrations of both redox oxidation states, while the $^{19}$F spin density contrast in Fig. 2e also reports on these concentrations due to the different relaxation times as well as the different numbers of anions associated with each cobalt redox state. While both nuclei are viable for imaging, in the majority of this work we have focused on $^1$H due to its higher sensitivity.

The strong dependence of the MPN $^1$H $T_1$ relaxation times on the local concentration of the cobalt redox species, particularly the paramagnetic $Co^{2+}$ state, is clear from Fig. 2d. However, these relaxation times are also temperature dependent (Fig. 2f), and both the concentration and temperature can vary spatially throughout the electrolyte in a functioning thermocell. The dependence of $T_1$ on these two variables must therefore be disentangled in order to interpret the relaxation contrast in a working device. We achieve this by first measuring a $T_1$ relaxation map of the thermocell with the temperature gradient applied but the cell in open-circuit mode (i.e., electrodes remaining disconnected). An initial assumption is made that local variations in the concentration of the redox species are negligible at open-circuit voltage, where no external electron transfer occurs to drive the redox reactions at the electrodes. This allows the acquired $T_1$ map to be directly converted to a temperature map via the (independently measured) temperature variation of the $^1$H $T_1$ for the particular electrolyte used (in this case MPN containing 0.05 M $Co^{2+}(bpy)_3(TFSI)_2$ and 0.05 M $Co^{3+}(bpy)_3(TFSI)_3$). This

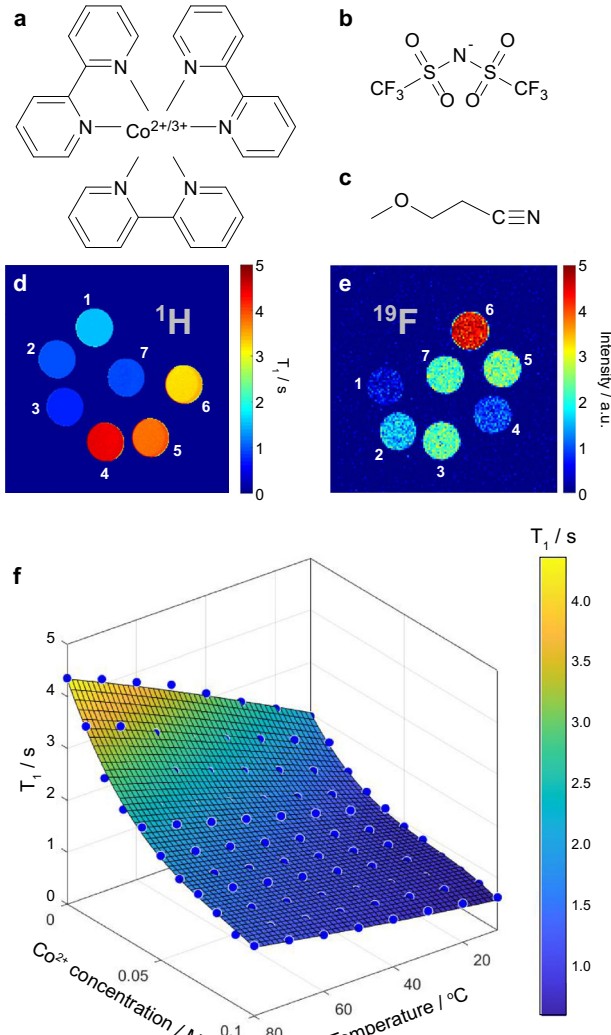

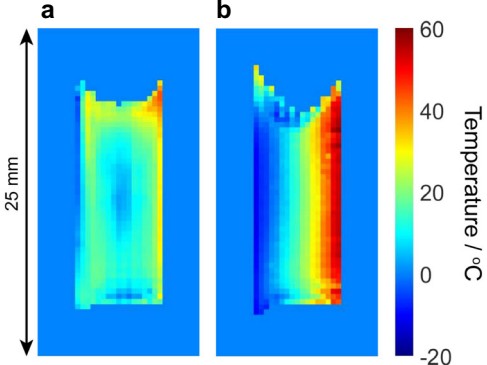

**Fig. 3 Temperature maps obtained from the vertical electrode cell.** The cells contained MPN with 0.05 M $Co^{2+/3+}(bpy)_3(TFSI)_{2/3}$ in **a** liquid and **b** gel form. In both cases, the target electrode temperatures were −30 °C (cold/left) and 60 °C (hot/right). The uncertainty in temperature for each pixel in these and subsequent images is ±2.5 °C.

**Fig. 2 The electrolyte system studied and $T_1$ relaxation data used for image calibrations.** Molecular structures of **a** the tris(bipyridyl)cobalt cation $Co(bpy)_3$, **b** the bis(trifluoromethanesulfonyl)-imide anion (TFSI) and **c** the 3-methoxypropionitrile (MPN) solvent. **d**, **e** Axial slice $^1H$ $T_1$ and $^{19}F$ spin density contrast images obtained from electrolyte samples in 5 mm diameter NMR tubes at 20 °C (MPN with (1) 0.05 M $Co^{2+}(bpy)_3(TFSI)_2$, (2) 0.10 M $Co^{2+}(bpy)_3(TFSI)_2$, (3) 0.15 M $Co^{2+}(bpy)_3(TFSI)_2$, (4) 0.05 M $Co^{3+}(bpy)_3(TFSI)_3$, (5) 0.10 M $Co^{3+}(bpy)_3(TFSI)_3$, (6) 0.15 M $Co^{3+}(bpy)_3(TFSI)_3$ and (7) both 0.05 M $Co^{2+}(bpy)_3(TFSI)_2$ and 0.05 M $Co^{3+}(bpy)_3(TFSI)_3$). **f** Plot of $^1H$ $T_1$ relaxation times (blue circles) of the MPN protons as a function of temperature and $Co^{2+}$ concentration, measured from electrolytes with a total $Co(bpy)_3$ concentration of 0.10 M. The surface is a best fit to a polynomial function obtained using MATLAB (see Supplementary Information for details).

requires the temperature map to be recorded as quickly as possible after establishing the thermal gradient in order to minimise the effects of thermophoresis, which can occur over a time scale of hours (vide infra). Subsequently, an external load resistance is applied and the cell begins to operate, with both the current and potential continuously measured via the potentiostat. A second assumption is then made that the resulting redox reactions, accompanying mass transfer processes such as diffusion and electrophoresis, and other effects such as Joule heating of the electrodes, do not significantly alter the temperature distribution within the electrolyte, which remains dominated by conductive and convective heat transfer between the electrodes.

This allows the temperature map obtained from the unconnected cell to be used as a restraint in calculating the local concentrations of the redox species from subsequent $T_1$ maps acquired from the functioning cell. This conversion is achieved using the polynomial function obtained from the surface fit of the $T_1$ data in Fig. 2f (see Supplementary Information for more details), which were acquired using separate calibration measurements carried out on individual samples prepared with different $Co^{2+}$ and $Co^{3+}$ concentrations. We note also that various other assumptions inherent to operando MRI studies of electrochemical devices also apply, for example that Lorentz forces on the electrolyte ions and eddy currents on the electrodes play a minimal role in the device functionality.

**Temperature maps and effects of convection.** Figure 3 shows temperature maps obtained from the liquid and gel electrolytes within the vertical electrode cell under a target temperature differential of 90 °C (although a narrower actual temperature difference across the electrolyte within the device is observed in both cases and this is attributed to the thermal gradient across the electrode itself). In these and all subsequent images presented, the pixel dimensions are $0.39 \times 0.39$ mm and the image dimensions are $12.5 \times 25$ mm. In the liquid electrolyte (Fig. 3a), a fairly uniform temperature of around 20 °C across the majority of the cell is observed, with a cooler region visible in the centre of the electrolyte and cold and hot layers visible close to the electrode surfaces. A hot region is also seen at the top of the hot electrode, possibly trapped by the meniscus of the electrolyte. The relatively uniform temperature across the majority of the electrolyte indicates rapid mixing due to thermal convection, and the cooler central region is consistent with modelling predictions of convective flow in a vertical electrode thermocell[17].

In the gel electrolyte under the same conditions (Fig. 3b), the effects of convection on the temperature distribution are absent as a result of the polymer matrix and a more linear temperature gradient is observed across the cell. These images provide a stark illustration of the role that convection can play in determining the temperature distribution within a thermocell, and the ability of gel electrolytes to maintain a large temperature differential by eliminating convective flow.

In this work, we have focused on 2D single-slice imaging in a vertical plane perpendicular to the electrodes and parallel to the thermal gradient, as this plane is expected to show the largest variations in both temperature and redox concentrations (i.e., along the directions of gravity, applied temperature gradient, and

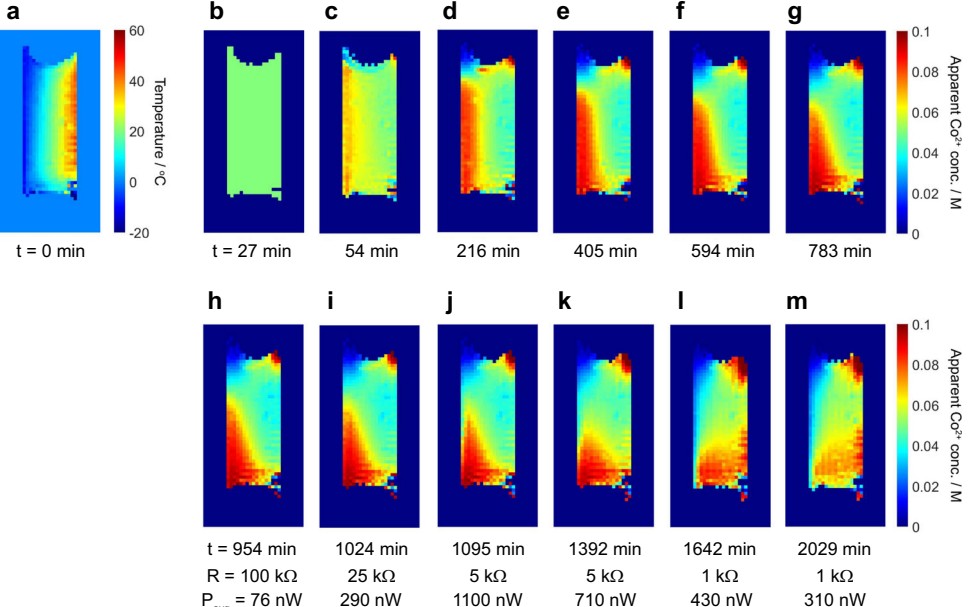

**Fig. 4 Imaging competing mass transfer effects in the vertical electrode cell. a** Temperature map and **b–m** apparent $Co^{2+}$ concentration maps obtained from the vertical electrode cell with the MPN with 0.05 M $Co^{2+/3+}(bpy)_3(TFSI)_{2/3}$ gel electrolyte at times $t$ indicated. Images **a–g** were obtained from an unconnected (open circuit) cell. Images **h–m** were obtained after connecting the load resistances $R$ as indicated. Average powers $P_{avg}$ drawn from the cell during the acquisition of each image are also shown. The uncertainty in concentration for each pixel in these and subsequent images is ±0.008 M. Plots of cell current and voltage versus time for these experiments are provided in Supplementary Fig. 11.

potential). However, it should be noted that the image slice can be set to any thickness, position and orientation. Additional temperature maps obtained from the liquid electrolyte under various applied temperature differentials, including with different image slice orientations (parallel and perpendicular to the electrodes), are shown in Supplementary Figs. 7–10. These images show some minor variation in the electrolyte temperature in the plane parallel to the electrodes, vertically due to convection, but also horizontally due to the temperature of the electrode surfaces not being perfectly uniform, as well as the effects of the interior side walls of the electrolyte chamber.

**Concentration maps and competing mass transfer effects**. For the vertical electrode cell with the gel electrolyte, a temperature map (Fig. 4a) and subsequent time-resolved series of concentration maps at open-circuit voltage and then with different applied load resistances (Fig. 4b–m) were acquired. Each image took ~27 min to obtain and the times quoted are the completion times for the acquisition of each image. First, $Co^{2+}$ concentration maps were obtained from the cell in its open-circuit state to observe the stability of the electrolyte over time under the effects of the applied thermal gradient (Fig. 4b–g). The initially uniform $Co^{2+}$ concentration across the whole electrolyte can be seen to change over a period of several hours, with an apparent clustering of $Co^{2+}$ species on the cold side of the cell (Fig. 4d) that eventually begins to sink to the bottom of the electrolyte (Fig. 4e–g). These observed changes are due to thermophoresis, whereby the cobalt species (of both oxidation states) migrate towards the cold side of the cell, gradually increasing the local electrolyte density and causing it to sink. This process coincided with a small reduction in the cell voltage (Supplementary Fig. 11). Note that this process results in a departure of the local net cobalt concentration from 0.10 M, and so the calibration data in Fig. 2f no longer applies, hence we have labelled the concentration scale in Fig. 4 as "apparent" $Co^{2+}$ concentration. The thermophoresis effect therefore limits the accurate quantification of the redox species

concentrations when this effect is dominant. Nonetheless, although these particular images are not truly quantitative, the contrast reliably illustrates the effects of thermophoresis and the time evolution of the spatial distribution of the cobalt species within the cell.

The thermophoresis process in Fig. 4b–d can be seen to occur over a time scale of hundreds of minutes, in agreement with previous experimental observations of the Soret effect[41]. This effect is mediated by the diffusion of the ions, which we have measured by pulsed-field gradient NMR to be on the order of $10^{-10}$–$10^{-9}$ $m^2 s^{-1}$ in the temperature range 20–80 °C. In the simplest case, the root-mean-square distance $L_{RMS}$ travelled by a molecule with a self-diffusion coefficient $D$ in time $t$ is given by the expression $L_{RMS} = (6Dt)^{0.5}$. Based on the measured diffusion coefficients for the $Co(bpy)_3$ species, $L_{RMS}$ is on the order of 2–6 mm for $t = 100$ min, length and time scales that agree well with our data.

In a functioning device, the effects of thermophoresis must be overcome by other mass transfer effects such as diffusion in order for the $Co^{3+}$ species to reach the hot electrode and undergo reduction to $Co^{2+}$. This can be observed in Fig. 4h–m, in which the cell was connected to a variable load resistance. Over time, and as the applied load resistance is reduced, the apparent $Co^{2+}$ concentration is seen to increase near the hot electrode, while a layer of reduced apparent $Co^{2+}$ concentration (i.e., increased $Co^{3+}$ concentration) grows at the surface of the cold electrode as expected. We note that this mass transport effect will also be mediated by ion diffusion and the length and time scale for the changes observed in the images in Fig. 4h–m also agree well with the measured diffusion coefficients of the $Co(bpy)_3$ species as mentioned above.

Subsequently, a fresh gel electrolyte was placed into the vertical electrode cell and a new series of concentration maps were generated, this time with the load resistances connected immediately after recording the temperature map, thereby minimising the effects of thermophoresis. The temperature map and an example $Co^{2+}$ concentration map are shown in Fig. 5a

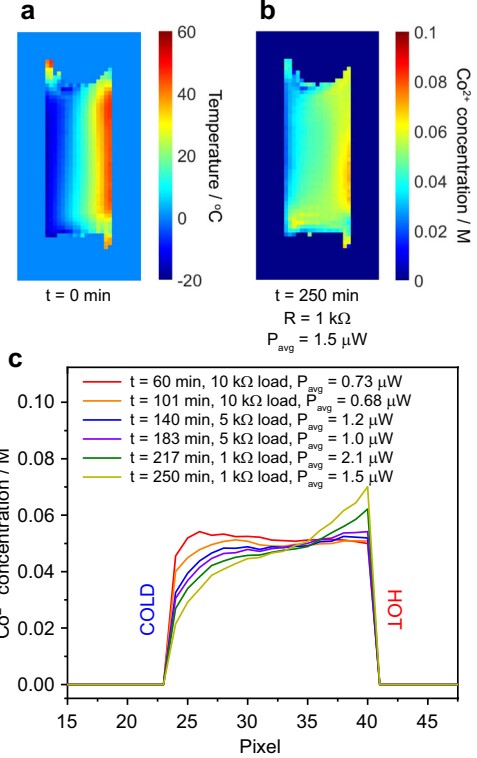

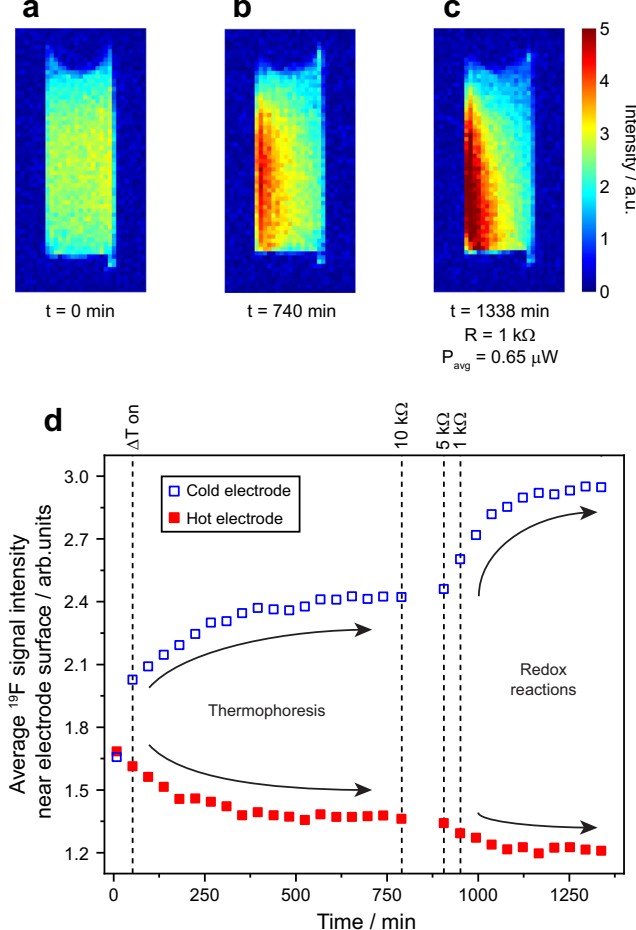

**Fig. 5 Build-up of concentration gradients in the vertical electrode cell. a** Temperature map and **b** $Co^{2+}$ concentration map obtained from the vertical electrode cell with the gel electrolyte. **c** One-dimensional $Co^{2+}$ concentration profiles across the centre of the cell, extracted from images obtained at the times $t$ and under the load resistances $R$ indicated (the yellow plot in (**c**) was taken from the image in (**b**)). Average powers $P_{avg}$ drawn from the cells during the acquisition of each image are also quoted. The full series of images are provided in Supplementary Fig. 16, along with plots of cell current and voltage versus time for these experiments (Supplementary Fig. 12).

and b, respectively, while the evolution of the $Co^{2+}$ concentration gradient across the centre of the cell as a function of time and load resistance is shown in Fig. 5c. As the effects of thermophoresis are minimal in this case, these $Co^{2+}$ concentration data are quantitative, and can be correlated with the variations in measured cell potential and current (Supplementary Figs. 12 and 16). The $Co^{2+}$ concentration gradient evolves from close to zero in the initial measurement to a sigmoidal-like profile under the 1 kΩ load resistance at $t = 250$ min. This is a departure from the linear concentration gradients predicted by Sokirko[16], and may be due to a number of factors not accounted for in the modelling such as the effects of the side walls of the electrolyte chamber, spatial variations in temperature across the surface of each electrode, and the fact that this system may not yet have reached the steady state. The concentration map in Fig. 5b also shows that the concentration profile varies vertically across the cell. These temperature and concentration maps therefore provide valuable experimental data to which modelling results may be compared.

$^{19}$F spin density images of the gel electrolyte in the vertical electrode cell are shown in Fig. 6. These images report on the local concentrations of the TFSI anions, which due to ion clustering will in turn depend on the local concentrations of the cobalt redox cations. $^{19}$F image intensity arising from the PVDF component of the gel is negligible due to its much broader NMR line width. In the image obtained from the unconnected cell with

**Fig. 6 $^{19}$F spin density maps obtained from the vertical cell and changes in anion concentration at the electrode surfaces. a** $^{19}$F spin density image of the gel electrolyte with no temperature gradient or load resistance, **b** after 740 min with thermal gradient applied but no load resistance connected and **c** after 1338 min with a 1 kΩ load resistance. In **b** and **c**, the cold electrode is on the left. **d** Time evolution of the $^{19}$F signal intensities at the surface of each electrode (average extracted from the four columns of pixels closest to each electrode, corresponding to a layer ~1.5-mm thick). The thermal gradient $\Delta T$ was switched on and the load resistances were applied at the times indicated by the dashed lines. Plots of cell current and voltage versus time for these experiments are provided in Supplementary Fig. 13.

no temperature gradient applied (Fig. 6a), the spin density is fairly uniform across the electrolyte as expected. The same thermal gradient as for the $^{1}$H imaging experiments discussed above was subsequently applied and after 740 min a clear increase in $^{19}$F spin density is observed on the cold (left) side of the cell (Fig. 6b), consistent with the migration of the anions to the cold side due to thermophoresis. Subsequently, a series of load resistances were applied and the image in Fig. 6c was acquired, showing a further increase in $^{19}$F spin density on the cold side. The evolutions in image intensities at the electrode surfaces during this series of experiments are plotted in Fig. 6d. Initially, the TFSI concentration increases at the cold electrode and decreases at the hot electrode due to thermophoresis. When the load resistances are connected, oxidation ($Co^{2+} \rightarrow Co^{3+}$) occurs at the surface of the cold electrode, necessitating a further local increase in TFSI concentration to balance the charge, while the opposite process occurs at the hot electrode. These processes are accompanied by significant decreases in the cell current and

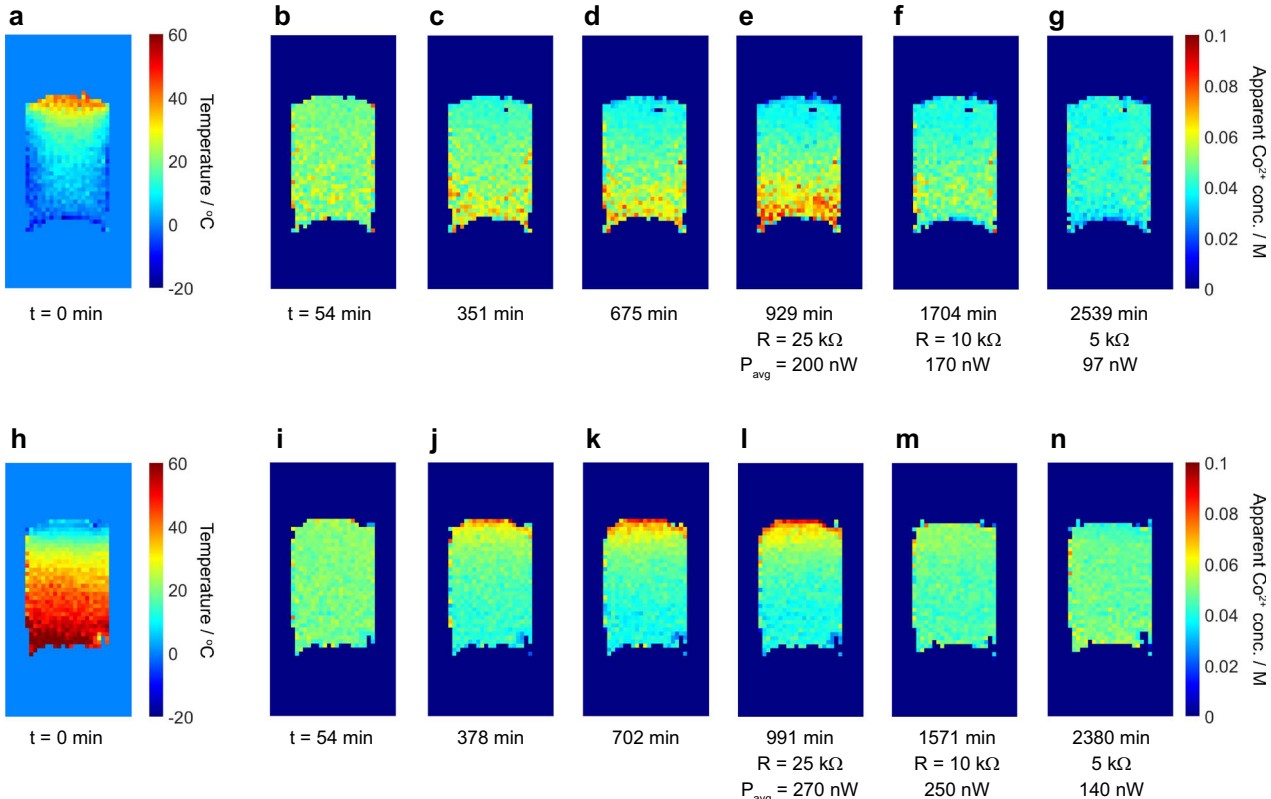

**Fig. 7 Competing mass transfer effects in the horizontal electrode cells.** Temperature and apparent $Co^{2+}$ concentration maps for the horizontal electrode cells in the hot-above-cold (**a–g**) and cold-above-hot (**h–n**) configurations. In both cases, the initial connection to an external load resistance was made at $t \approx 700$ min. Plots of cell current and voltage versus time for these experiments are provided in Supplementary Figs. 14 and 15.

potential (Supplementary Fig. 13) as the system moves towards equilibrium. The observation of larger TFSI concentration changes near the cold electrode (when one might expect the changes at each electrode to be equal and opposite) is consistent with the predictions of modelling and is due to the slower rates of diffusion on the cold side of the cell, which leads to more pronounced diffusive layers at the cold electrode surface (see for example Fig. 2c in ref. [17]).

Finally, a series of $^1H$ imaging experiments were carried out on the gel electrolyte in the horizontal electrode cell with both hot-above-cold and cold-above-hot configurations (Fig. 7). The effects of such orientations on thermocell performance, and particularly the advantage of the cold-above-hot set up in promoting mass transport via thermal convection, have been previously established[21,22]. Figure 7a and h shows the acquired temperature maps for these configurations, and the temperature gradient is seen to be non-linear and steeper at the top of the cell in both cases. We note that the temperature distribution in the hot-above-cold cell (Fig. 7a) shows more variation in the horizontal direction than that in the cold-above-hot cell (Fig. 7h). We attribute this to differences in heat loss to the PEEK side walls that result from the different gas flow pathways inside the cell for these two set ups (shown schematically in Fig. 1b and c and a result of the restrictions on the gas inlet placements). We also note some distortions to the images which make the top and bottom surfaces appear slightly curved (most noticeable for the hot-above-cold images in Fig. 7a–g), and this is attributed to magnetic field inhomogeneities close to the electrodes.

For both the hot-above-cold and cold-above-hot configurations, the effects of thermophoresis can be observed with an increase in the apparent $Co^{2+}$ concentration appearing over several hours at the cold region of the electrolyte before any load

resistance is connected (Fig. 7b–g and i–n). Upon connecting a load resistance, the concentration gradient is gradually seen to reverse as other mass transfer processes driven by the redox reactions begin to dominate. Average powers $P_{avg}$ drawn from the cells during each imaging experiment are around 50% higher for the cold-above-hot configuration (cell currents and voltages are shown in Supplementary Figs. 14 and 15), indicating a more efficient mass transfer between the electrodes. This could be due to the differences in the thermal gradients (Fig. 7a and h), some slow or small-scale thermal convection within the gel, and/or the sinking of the more dense regions of electrolyte with higher $Co^{3+}$ concentration that are formed at the upper cold electrode, neither of which would occur for the opposite orientation.

In summary, we have shown that operando MRI can be used to obtain quantitative temperature and concentration maps from working thermocells, providing valuable experimental data on the different inter-related processes occurring in these devices that can be correlated with both modelling results and electrochemical measurements. The inherent paramagnetism of the $Co^{2+}$ redox state led to significant $^1H$ $T_1$ relaxation contrast, making thermocells particularly well-suited for characterisation by this technique, and $^{19}F$ spin density imaging was also demonstrated as a viable alternative. We have observed thermophoresis occurring inside a gel electrolyte in an unconnected cell, and have shown that other mass transfer processes induced by the redox reactions when a load resistance is applied are dominant over this process and reverse its effects. The image acquisition times used were short relative to the time scale of the observed concentration changes, allowing them to be monitored as a function of time and applied load resistance. The $^1H$ relaxation-based MRI approach we present in this work should be applicable to any electrolyte system containing a protonated solvent molecule (e.g., water, or

organic solvents) and a redox couple where one of the redox states is paramagnetic to provide enhanced relaxation contrast. Spin density imaging, also demonstrated here using $^{19}F$, may be applicable to non-protonated systems, redox couples with no paramagnetic state, or solid or mixed-phase electrolytes provided they contain magnetic nuclei with sufficient sensitivity and suitable relaxation times. In this work, we have focused on 2D imaging, but multi-slice imaging or more advanced pulse sequences could be used to provide 3D images featuring spatial information in all directions. Other MRI modalities such as diffusion and velocity mapping will provide further insights into mass transfer processes such as electrophoresis and convection. The approaches reported herein may also be useful for studying other energy storage devices such as thermal batteries and capacitors.

## Methods

**Electrolyte preparation**. Reagents for the cobalt redox couple synthetic procedures were purchased and used as received from Sigma-Aldrich, 3 M, May & Baker Ltd, Emsure, and LiChrosolv. 3-Methoxypropionitrile (MPN) (98%) was purchased from Fluka and used as received. The redox couples $Co^{2+}(bpy)_3TFSI_2$ and $Co^{3+}(bpy)_3TFSI_2$ (where bpy = 2,2-bipyridyl and TFSI = bis(trifluoromethanesulfonyl)imide) were synthesised following a previously published procedure[23]. Polyvinylidene difluoride (PVDF) powder (KF850, molecular weight = $3 \times 10^5$) was purchased from Kureha Chemicals, Japan, and used as received, and gel electrolytes were prepared by incorporating 5 wt% PVDF into the liquid electrolytes.

**$^1H$ relaxation calibration measurements**. A series of electrolytes with varying fractions of $Co^{2+}(bpy)_3TFSI_2$ and $Co^{3+}(bpy)_3TFSI_2$ but a total $Co(bpy)_3$ concentration of 0.10 M were prepared in both liquid and gel forms. $^1H$ $T_1$ relaxation times were measured over a range of sample temperatures from the MPN signals using an 11.7 T Bruker Avance III standard bore spectrometer, a 5 mm HX solution-state probe, and a saturation recovery pulse sequence. The resulting data were fitted to a polynomial surface function (Fig. 2) and the resulting parameters of this fit are provided in the Supplementary Information.

**Diffusion measurements**. Self-diffusion coefficients of the $Co(bpy)_3$ species were measured using $^1H$ pulsed-field gradient NMR on a Bruker Avance III 7.05 T spectrometer with a Diff50 Z-gradient probe and using a stimulated echo pulse sequence. Gradient strengths were varied up to 3000 Gauss/cm and diffusion coefficients were extracted by fitting the resulting signal attenuation curves to the Stejskal-Tanner equation.

**Thermocell designs**. The cells designs were developed in SolidWorks and are available upon request. Further images of the cells are provided in the Supplementary Information. They were machined from polyether ether ketone (PEEK) and measure 25 mm in diameter (30 mm diameter cap) and 100 mm in length. The wider cap at the top allows the cell to rest in the correct position within the imaging probe. Cell components are connected using PEEK screws with butyl rubber gaskets (1 mm thickness) to prevent electrolyte leakage. 0.25-mm-thick platinum electrodes were also held in place by butyl rubber gaskets. The surface area of each electrode exposed to the electrolyte was ~1.4 cm$^2$ for the vertical electrode cell and 0.4 cm$^2$ for the horizontal electrode cell. For the vertical electrode cell design, the electrolyte is inserted from above by removing the top cap of the cell. For the horizontal electrode cells, an inlet on the side of the cell allows the electrolyte to be injected into the cavity before being sealed with a PEEK screw.

**Operando MRI experimental set up**. The cells were placed inside the 25 mm RF coil of a Bruker Micro2.5 microimaging probe. Separate exchangeable MICWB40 birdcage RF coils were used for $^1H$ and $^{19}F$ imaging experiments, and the imaging probe was itself placed inside a triple-axis (XYZ) gradient system housed within the bore of the superconducting magnet. The Z gradient was oriented vertically (parallel to the applied magnetic field), while combinations of X and Y gradients were used to align the transverse gradients relative to the cell orientation. The hot and cold gas flows were controlled using two independent Bruker BCU II chiller units. Gas lines and electric cables were connected to the cell before placing the probe inside a Bruker 11.7 T wide-bore vertical superconducting magnet with an Avance III console and three GREAT (1/60) amplifiers capable of generating X, Y and Z field gradients of up to 1.5 Tm$^{-1}$. In the case of the vertical electrode configuration, the cell was aligned with the electrode surfaces parallel to the RF ($B_1$) field to avoid image distortions[30]. The electrode connection cables incorporated an RF filter at the top of the magnet to reduce noise in the acquired images, and were connected to an external potentiostat. A schematic illustration of the set-up is provided in Supplementary Fig. 5.

**Electrochemical measurements**. During all imaging experiments, the thermocell voltage and drawn current were continuously recorded using a BioLogic SP-150 (Science Instruments) potentiostat controlled using the EC-Lab software, which was also used to collect the data. This potentiostat was used to control the applied load resistance (constant load discharge), which was varied over the course of each experiment in such a way as to best illustrate variations in the images acquired. The load resistance values and timings used are indicated separately in each figure, along with average thermocell output powers $P_{avg}$ measured during the acquisition of each image. The raw voltage and current data measured as a function of time for all images shown are provided in Supplementary Figs. 11–15.

**$^1H$ relaxation mapping**. Imaging experiments were controlled using the Bruker ParaVision software (v6). The rapid acquisition with relaxation enhancement (RARE) imaging pulse sequence was used for all imaging experiments. Images were obtained with a RARE factor of 1 (i.e., one acquisition of signal per scan), a field of view of 25 × 25 mm and a k-space matrix dimension of 64 × 64, which translate to the image pixel dimensions of 0.39 × 0.39 mm after processing. Unless otherwise stated, the images acquired represent vertical slices positioned in the centre of the electrolyte and in a plane perpendicular to the electrodes, with slice thicknesses of 8 and 5 mm for the vertical and horizontal electrode configurations, respectively. Information represented by each pixel therefore represents an average value for the electrolyte across these distances (perpendicular to the image plane). A series of $T_1$-weighted images were obtained using 15 experiments with repetition times logarithmically spaced between 100 and 6000 ms. These different repetition times sample the longitudinal magnetisation of the $^1H$ spins as they undergo relaxation. The minimum possible echo time of 5.08 ms was used to minimise the effects of transverse relaxation, and the total acquisition time for each $T_1$ map (i.e., all 15 $T_1$-weighted images) was 27 min. Signal intensities below a certain threshold (15–25%) of the maximum value (i.e., noise outside of the electrolyte region) were removed. The $T_1$ maps were then reconstructed from this series of 15 images using the Prospa software (Magritek, New Zealand) by fitting the signal intensity variation of each pixel using the function $S(t) = S_{max}(1 - e^{\frac{-t}{T_1}})$, where $S(t)$ is the pixel signal intensity for repetition time $t$ and $S_{max}$ is the maximum signal intensity for the pixel.

**$^{19}F$ spin density imaging**. The $^{19}F$ spin density images (i.e., where pixel brightness reflects the local concentration of fluorine nuclei) were also acquired with the RARE pulse sequence, with a repetition time of 20 s to avoid $T_1$ contrast, and a RARE factor of 1 and a minimum echo time of 5.08 ms to minimise $T_2$ relaxation contrast. The slice thickness, field of view and matrix size were identical to the $^1H$ imaging, but due to the weaker signal strength of $^{19}F$ (due to its lower resonance frequency as well as lower concentration) additional signal averaging of 2 was employed resulting in an acquisition time of 43 min per image.

**Data processing**. The $^1H$ $T_1$ maps were converted to either temperature or $Co^{2+}$ concentration maps using the relaxation calibration data measured separately (see Fig. 2f and Supplementary Information for full expression) and following the general procedure outlined in the main text via a home-written MATLAB code. All imaging figures were generated using MATLAB and cropped in Adobe Illustrator, while data plots were generated using Origin.

## Data availability
The imaging and electrochemistry data generated in this study have been deposited in the open access zenodo database (URL: https://zenodo.org/record/5565584#.YWY_lRqb670, https://doi.org/10.5281/zenodo.5565584)[42].

## Code availability
The MATLAB code used for image processing in this study has been deposited in the open access zenodo database (URL: https://zenodo.org/record/5565584#.YWY_lRqb670, https://doi.org/10.5281/zenodo.5565584)[42].

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

## Acknowledgements

Prof. Maria Forsyth is thanked for her valuable comments on this work. We are grateful to Dr. Abuzar Taheri and Dr. Ruhamah Yunis for providing the electrolyte samples and to Mr. John Taylor for machining the cells. The Australian Research Council is acknowledged for funding the microimaging facility via grant LE110100141 (Prof Forsyth), and for supporting this research through the ARC Centre of Excellence for Electromaterials Science (ACES).

## Author contributions

I.E.G. and L.A.O. conceived the project. I.E.G. designed the cells, carried out the experiments and data analysis, led the data interpretation and assisted with the paper writing. J.M.P. assisted with the project execution, data interpretation and paper writing. L.A.O. assisted with the project execution and data interpretation, and wrote the paper.

## Competing interests

The authors declare no competing interests.
