## [Peer Review File · Nature Communications]

REVIEWER COMMENTS

Reviewer #1 (Remarks to the Author):

The manuscript "Operando MRI for quantitative mapping of temperature and redox species concentrations in thermo-electrochemical cells" used MRI approach to provide quantitative spatial maps of electrolyte temperature and redox ion concentrations in functioning thermo-electrochemical cells. The work is very interesting and well presented, it could greatly contribute to the understanding of the mechanism in thermocell and thermoelectric related devices, which is a very challenging topic. I recommend this paper to be accepted by Nature Communications after explaining a few of the reviewer's concerning:

1. From the reviewer's point of view, the operation principle of MRI is not very well known for scientists in different fields, especially new in research field of thermocells (which is one of the highlight of this work). The author introduced the principal of MRI quite clear; however, it would be helpful if the author could provide more detailed information of the operation of MRI and the combination with thermocell in the illustration figures and discussion, e.g. how does the xyz gradient system works in terms of mapping the thermocell? what is the dimension of the electrodes of the thermocell? Since the chamber of the thermocell has 3 dimension, the MRI mapping is probing the surface of the corss-section of the cell or the average distribution or somewhere else? These are just listed as examples, I believe this work will attract more audience if the MRI method is presented more pedagogically.
2. In Figure 4 a-g, the authors studied the mass transport and aggregation of Co ions, I wonder how is the evolution of the thermovoltage of the cell in the process? Similar for the following figures showing the redistribution of Co ions after connecting load resistance, how is the output current during these processes? It would be great if the author could correlated the MRI image to the open circuit thermovoltage and output current data.
3. In the paper, all the temperature characterization was done by direct MRI mapping, which is very convenient and provide spacial distribution. I am slightly concerning about the assumption that the thermodiffusion of mass will be delayed compare to the temperature gradient. From some reference, the depletion of ions/species could happen immediately (Eur. Phys. J. E (2017) 40: 4, Phys. Rev. E 99, 042136, 2019, Nano Lett. 2017, 17, 3145–3151). The author mentioned that it took 25 mins to finish the mapping, during this time, how is the thermodiffusion? Since the results are quite relied on the temperature mapping, I wonder if the author could map the temperature via other methods to exclude the contribution from thermodiffusion? Or maybe verify that in the studied system, the thermodiffusion is slow compare to the establishing of temperature gradient. One way could be to track the open circuit voltage after applying temperature, if there is no thermovoltage developed during the temperature mapping period, then the assumption can be confirmed.
4. The last suggestion from the reviewer, the MRI method was demonstrated very useful in the studied system in this work, could the author provide some description regarding the applicable materials that is suitable to use MRI to investigate? And the limitations of MRI, which type of material and system will not be possible? It might be useful discussion at the end of the manuscript to offer the audience the scope of this method.

Reviewer #2 (Remarks to the Author):

The authors use MRI to image the interior of a thermo-electrochemical cell, allowing for species concentration and temperature to be measured in the electrolyte. The study considers a liquid and gel electrolyte, along with the applied temperature difference in the thermocell set in different configurations (e.g. one vertical and two horizontal layouts). The experimental results provide quantitative data previously available only through numerical modeling. It is suggested the paper is published after revision.

1. The authors mention Salazar's model was restricted to two dimensions. The results here are similarly in one plane, showing in effect 2D results.
 - a. Can the authors show results in additional planes to highlight additional spatial information (perpendicular to current results or parallel to the electrode plane)?
 - b. Can the authors comment on the effect of slice thicknesses of 8 mm and 5 mm for the two configurations in terms of the temperature and concentration results? To what extent does this volume average influence the presented results in terms of accuracy or losing potentially valuable 3D information? Could thinner slices be used in the future (while still maintaining acceptable temporal resolution)?
2. The authors note the issue of apparent concentration that occurs due to thermophoresis. Can this issue be overcome to provide true concentration values, and to what extent does this limit the method and comparison with models?
3. On the temperature measurements and results:
 - a. How was the steady state assumption of the temperature profile assessed both before experiments began and during the experiments? Does the temperature profile not change during operation in the unconnected or connected cells?
 - b. What is the temperature range of the results in figure S7a – this should be isothermal at 25°C but it appears there is some spatial variation in temperature. Is this variation physically occurring or is it due to measurement?
 - c. What is the accuracy of the temperature results in any given pixel? What is the accuracy of the heating and cooling streams in terms of keeping an isothermal boundary at the electrode surface?
 - d. Can the authors explain more why the desired/set temperature gradient was not measured in the results (i.e. -20°C in exp vs. -30°C desired)?
 - e. Have the authors considered running ex-situ experiments with embedded thermocouples to verify the MRI calibration procedure (or considered other methods of verification)?
 - f. In general, the temperature maps for vertical and cold-above-hot show a 1D profile in the temperature distribution while the hot-above-cold results show a stronger 2D variation in temperature (i.e. -20°C at the bottom boundary but also 20°C along the electrolyte container walls). Why is this the case?
4. The overall measurement accuracy of the method should be commented on for the concentrations and temperature. Can the authors further comment on expected repeatability of the method?
5. What were the number of averages used for the different sequences? 1H relaxation maps took 27 minutes, yet TR was varied from 100-6000ms, which will change the duration of the scan. The 19F images were acquired with a TR of 20s and 'additional signal averaging was employed resulting in an acquisition time of 43 minutes.' At TR=20s, this would allow for only 2 averages, less than that implied by the 27 min scan time of the 1H images.
6. Why is only 250 minutes considered for Figure 5 while the results in Figure 4I&M imply transient behavior at that resistance value over that timescale?
7. The authors note thermal convection in the liquid cell in Figure 3a. Was this measured by obtaining the velocity profile, inferred from the temperature results, or consistent with previous modeling?
8. On line 195, it is noted that convection is prevented in the gel electrolyte. However, the authors note small scale thermal convection is possible on line 291. This difference should be clarified.

Reviewer #3 (Remarks to the Author):

This is a very interesting system and has produced some nice measurements. My main criticism is it's over-engineered, and I find it highly unlikely anyone else in the world would adopt this form of measurement! It's also unfortunate that this particular form of measurement cannot be extended to the vast majority of electrolytes actually employed in thermocells. This does effect the impact of this work to a degree.

Fundamentally, it's a nice idea, nice setup and has produced some nice results. I feel it merits publication, but have the below comments:

(1) The manuscript undersells the power of IR cameras (ref. 25). And it also doesn't mention other work involving detailed characterisation using IR cameras, such as reference 11. This makes the manuscript a bit misleading; the authors should expand upon prior work (such as ref 11) and instead point out what this technique can do that the IR camera cannot (e.g. Soret). Very clearly external IR imaging can strongly reflect the interior of the cell.

(2) The same extends to modelling; I believe only 2 modelling studies were mentioned, but ignores modelling such as Ref 11. A very quick search also turned up Wu et al. (2017, ELECTROCHIMICA ACTA, 2017, 225, 482-492, <https://doi.org/10.1016/j.electacta.2016.12.152>) and in the ESI of Im et al (NANO RESEARCH, 2014, 7, 443-452, <https://link.springer.com/article/10.1007/s12274-014-0410-6>); I am sure there are others.

(3) The thermocell measurements conditions are poorly described; I cannot see any relevant data from the actual measurements, there's no description in the Methods, and the manuscript only contains a cryptic "Subsequently, an external load resistance is applied and the cell begins to operate, with both the current and potential continuously measured via the potentiostat." statement.

(4) Since potential was measured, the Nernst equation can be used to model the predicted ratio's of Co(II/III) at the electrode surfaces (cf. Buckingham et al., SUS ENERGY FUELS, 2020, 4, 3388-3399, <https://pubs.rsc.org/en/content/articlelanding/2020/se/d0se00440e#!divAbstract>). Do the values measured and plotted in Fig 5(c) match with the calculated values? Presumably any difference is due to either the effect of ratio upon the seebeck coefficient, and/or Soret.

(5) Various areas of the R&D are not compared vs prior literature and are thus treated as novel, e.g. on page 8 it's stated that gelling the electrolyte switched off convection (something well understood and previously reported; the novelty here is imaging in), on page 9 the 'sinking effect' is noted (that's been mentioned before in a lot of the earlier fundamental work, e.g. by Ikeshoji [1987, Bull. Chem. Soc. Jpn., 60, 1505-1514]) and the hot-over-cold / cold-over-hot is introduced without any context. In particular, the average reader will not understand how this orientation study is actually a 'failure' of the experiment to match expectations because genuinely gelled electrolytes should not have been effected by orientation - and either (a) this technique is visualising things which are hard to conceive and observe in the macro state, or (b) the gel melted!

Response to reviewer comments for manuscript NCOMMS-21-21865-T

Below, we provide detailed responses to all of the reviewer concerns (reviewer comments in red and our responses in black), and the various revisions and additions to the manuscript and its supplementary information are highlighted in yellow in those documents.

Reviewer #1 (Remarks to the Author):

The manuscript “Operando MRI for quantitative mapping of temperature and redox species concentrations in thermo-electrochemical cells” used MRI approach to provide quantitative spatial maps of electrolyte temperature and redox ion concentrations in functioning thermo-electrochemical cells. The work is very interesting and well presented, it could greatly contribute to the understanding of the mechanism in thermocell and thermoelectric related devices, which is a very challenging topic. I recommend this paper to be accepted by Nature Communications after explaining a few of the reviewer’s concerning:

1. From the reviewer’s point of view, the operation principle of MRI is not very well known for scientists in different fields, especially new in research field of thermocells (which is one of the highlight of this work). The author introduced the principal of MRI quite clear; however, it would be helpful if the author could provide more detailed information of the operation of MRI and the combination with thermocell in the illustration figures and discussion, e.g. how does the xyz gradient system works in terms of mapping the thermocell? what is the dimension of the electrodes of the thermocell? Since the chamber of the thermocell has 3 dimension, the MRI mapping is probing the surface of the cross-section of the cell or the average distribution or somewhere else? These are just listed as examples, I believe this work will attract more audience if the MRI method is presented more pedagogically.

We appreciate the reviewer’s point that MRI is not a well-known technique among the community of thermocell researchers and we agree that some more basic information about this approach is merited. To address this, we have made the following additions to the manuscript:

(1) Highlighted two informative review articles on operando MRI techniques applied to electrochemical devices in the introduction section. These two articles cover many of the important practical considerations around use of this technique.

(2) Expanded the methods section in numerous places to provide further explanation and background information on the MRI set up, imaging protocols and data analysis. We believe that the technical details are now much more understandable to a non-expert, and the reviewer’s specific questions about the gradient system and how the images represent a cross section of the electrolyte are now addressed.

Regarding the reviewer’s question about the electrode dimensions, the thickness of the platinum plates and the surface areas in contact with the electrolyte for the horizontal and vertical electrode set ups were already mentioned in the methods section (thermocell designs section).

2. In Figure 4 a-g, the authors studied the mass transport and aggregation of Co ions, I wonder how is the evolution of the thermovoltage of the cell in the process? Similar for the following figures showing the redistribution of Co ions after connecting load resistance, how is the output current during these

processes? It would be great if the author could correlate the MRI image to the open circuit thermovoltage and output current data.

Reviewer 3 also requested this data (see below). While we did provide the average power drawn from the cells during the acquisition of each of these images in the figures, we agree that the voltage and current data will be of interest to many readers. These data were measured during the image acquisitions and we have now incorporated the plots of these quantities (for all images shown) in the Supplementary Information (Figures S11 to S15).

3. In the paper, all the temperature characterization was done by direct MRI mapping, which is very convenient and provides spatial distribution. I am slightly concerned about the assumption that the thermodiffusion of mass will be delayed compared to the temperature gradient. From some references, the depletion of ions/species could happen immediately (Eur. Phys. J. E (2017) 40: 4, Phys. Rev. E 99, 042136, 2019, Nano Lett. 2017, 17, 3145–3151). The author mentioned that it took 25 mins to finish the mapping, during this time, how is the thermodiffusion? Since the results are quite relied on the temperature mapping, I wonder if the author could map the temperature via other methods to exclude the contribution from thermodiffusion? Or maybe verify that in the studied system, the thermodiffusion is slow compared to the establishing of temperature gradient. One way could be to track the open circuit voltage after applying temperature, if there is no thermovoltage developed during the temperature mapping period, then the assumption can be confirmed.

This is a very important point as it concerns not just our interpretation of the images but the entire approach we take in generating the temperature and concentration maps. Regarding the references mentioned by the reviewer:

(1) The Eur. Phys. J. E paper uses non-equilibrium molecular dynamics modelling to measure the Soret coefficient in a binary mixture under the application of a sudden local heating, and investigates the time scale for separation of the components due to the Soret effect. The two components were modelled as particles with the same diameter, no electric charge, and a mass difference ratio of 10, which is a significant departure from our system. The primary conclusion of the paper is that the Soret effect can be observed in a binary system immediately after a temperature differential is applied “in the vicinity of the hot layer [...], whereas separation in the bulk is slow”. A quantitative comparison of these modelling results with our experimental data is hampered by the dramatic differences in the model used and our system, as well as their use of reduced (dimensionless) Lennard-Jones temperature, time and distance units.

(2) The Phys. Rev. E paper presents a theoretical and numerical study of the Soret effect, this time considering ionic species, but again using a model that differs dramatically to our experiments (i.e., a temperature quench of just 0.3 K and an inter-electrode distance in the nm range). Again, this makes a quantitative comparison with our observations extremely difficult.

(3) The Nano Lett. Paper concerns a study of plasmonic nanohole array electrodes, including application in a thermoelectric device. The system dimensions are on the order of nm (176 nm diameters holes in a 30 nm thick film). This is a fundamentally different device design than the thermocells we have characterised. The Soret effect does not appear to be discussed in this paper.

We have found papers demonstrating a very slow Soret effect in liquids over mm/cm length scales. For example Tanner (Trans. Faraday Soc. 49 (1952) 611-619) defines the effect as “the slow development of a concentration gradient...” and mentions a time scale of “several 100 min” for the process occurring in an aqueous solution of KCl (a more fluid system than the electrolyte we study). Tanner studied this system using optical measurements and a vertical temperature gradient with cell dimensions and temperature differentials much closer to those in our work than the three papers

described above, and his quoted time scale of several hundred minutes agrees very well with our observations.

More fundamentally, one can consider the fact that the Soret effect is (over long distances) mediated by the diffusion of the ions in the solution. We have now measured the self-diffusion coefficients of the ions and solvent molecules in our system using pulsed field gradient NMR (data to be published separately). In both the liquid and gel electrolytes the diffusion coefficients of the $\text{Co}(\text{bpy})_3$ species are on the order of 10^{-10} to $10^{-9} \text{ m}^2\text{s}^{-1}$ in the temperature range 20 to 80 °C. In the simplest case, the root-mean-square distance L_{RMS} travelled by a molecule with a self-diffusion coefficient D in time t is given by the expression $L_{\text{RMS}} = (6Dt)^{0.5}$. Based on the measured diffusion coefficients for the $\text{Co}(\text{bpy})_3$ species, L_{RMS} is on the order of 2-6 mm for $t = 100$ min. This is in excellent agreement with Tanner's observations and with our data. Of course, the same principles apply to the other mass transport effects we observe after connecting our cells.

To clarify these important points in our manuscript, we have now incorporated the Tanner reference and the above calculation in the discussion of Figure 4. We thank the reviewer for prompting us to verify this.

4. The last suggestion from the reviewer, the MRI method was demonstrated very useful in the studied system in this work, could the author provide some description regarding the applicable materials that is suitable to use MRI to investigate? And the limitations of MRI, which type of material and system will not be possible? It might be useful discussion at the end of the manuscript to offer the audience the scope of this method.

We thank the reviewer for this suggestion. We have added the following text to the concluding paragraph to address this:

"The ^1H relaxation-based MRI approach we present in this work should be applicable to any electrolyte system containing a protonated solvent molecule (e.g., water, or organic solvents) and a redox couple where one of the redox states is paramagnetic to provide enhanced relaxation contrast. Spin density imaging, also demonstrated here using ^{19}F , may be applicable to non-protonated systems, redox couples with no paramagnetic state, or solid or mixed-phase electrolytes provided they contain magnetic nuclei with sufficient sensitivity and suitable relaxation times."

Reviewer #2 (Remarks to the Author):

The authors use MRI to image the interior of a thermo-electrochemical cell, allowing for species concentration and temperature to be measured in the electrolyte. The study considers a liquid and gel electrolyte, along with the applied temperature difference in the thermocell set in different configurations (e.g. one vertical and two horizontal layouts). The experimental results provide quantitative data previously available only through numerical modeling. It is suggested the paper is published after revision.

1. The authors mention Salazar's model was restricted to two dimensions. The results here are similarly in one plane, showing in effect 2D results.

a. Can the authors show results in additional planes to highlight additional spatial information (perpendicular to current results or parallel to the electrode plane)?

We thank the reviewer for this suggestion. This technique can indeed be used to acquire images in any orientation and it is certainly of interest to study variations in other directions even if these may be more subtle. We have now incorporated various examples of temperature maps corresponding to thin (1 mm thick) slices of the electrolyte at various orientations and positions in the Supplementary

Information (Figures S8 – S10) to illustrate that this is possible, and we have added the following paragraph to the manuscript to briefly discuss these:

“In this work we have focused on 2D single-slice imaging in a vertical plane perpendicular to the electrodes and parallel to the thermal gradient, as this plane is expected to show the largest variations in both temperature and redox concentrations (i.e., along the directions of gravity, applied temperature gradient, and potential). However, it should be noted that the image slice can be set to any thickness, position and orientation. Additional temperature maps obtained from the liquid electrolyte under various applied temperature differentials, including with different image slice orientations (parallel and perpendicular to the electrodes), are shown in the Supplementary Information (Figures S7 – S10). These images show some minor variation in the electrolyte temperature in the plane parallel to the electrodes, vertically due to convection, but also horizontally due to the temperature of the electrode surfaces not being perfectly uniform, as well as the effects of the interior side walls of the electrolyte chamber.”

Additionally we have added the following sentence to the concluding paragraph:

“In this work we have focused on 2D imaging, but multi-slice imaging or more advanced pulse sequences could be used to provide 3D images featuring spatial information in all directions.”

b. Can the authors comment on the effect of slice thicknesses of 8 mm and 5 mm for the two configurations in terms of the temperature and concentration results? To what extent does this volume average influence the presented results in terms of accuracy or losing potentially valuable 3D information? Could thinner slices be used in the future (while still maintaining acceptable temporal resolution)?

The relatively large slice thicknesses were selected to maximise the experimental sensitivity (i.e., signal strength). The slices provide an average over the slice thickness in the direction normal to the image plane, so the reviewer is correct that in this case spatial information in that direction is lost. We have also carried out multiple slice experiments with multiple, offset (thinner and non-overlapping) slices acquired. This can be done without sacrificing temporal resolution and can provide a more complete 3D picture of the entire electrolyte, however, the overall sensitivity is reduced and the images suffer from a poorer signal-to-noise ratio as a result. We have added some examples of such images to the Supplementary Information (Figures S9 and S10), but we have kept the focus of the main manuscript on the plane parallel to the directions of gravity, applied temperature gradient and potential, as these images showed the most significant and interesting variations in both temperature and concentration.

2. The authors note the issue of apparent concentration that occurs due to thermophoresis. Can this issue be overcome to provide true concentration values, and to what extent does this limit the method and comparison with models?

This is an inherent limitation to our approach, which relies on the assumption of a constant local net cobalt concentration throughout the electrolyte. At steady state operation this assumption is likely to be valid (see for example Figure 4 in reference [16]). However, when thermophoresis is dominant, the total local cobalt concentration will vary significantly, thereby introducing an additional variable to the function required to convert the ^1H relaxation (T_1) times to Co^{2+} concentrations, and it would be very challenging to solve such an equation reliably. We have now added the following sentence to the discussion of Figure 4 to clarify this limitation:

“The thermophoresis effect therefore limits the accurate quantification of the redox species concentrations when this effect is dominant.”

3. On the temperature measurements and results:

a. How was the steady state assumption of the temperature profile assessed both before experiments began and during the experiments? Does the temperature profile not change during operation in the unconnected or connected cells?

The reviewer is correct that the utility of this approach depends on the temperature distribution in the electrolyte remaining constant over time. Unfortunately the temperature maps cannot be “checked” by MRI at later stages during these experiments because they rely on a uniform concentration of the redox species. Once thermophoresis or other mass transport processes have caused appreciable concentration gradients, the MRI data can no longer be used to observe new temperature maps. Our verification of the accuracy of the temperature maps has therefore been limited to bench-top temperature measurements using in situ thermocouples placed at various locations within the cell. These have agreed well with our temperature images (to within a few °C) and show little variation once the device is operational.

We can also carry out a back-of-the-envelope calculation to determine what temperature changes might be expected in the electrolyte as a result of the power generation of the device, assuming that this power reflects the energetics associated with the processes occurring inside the cell during its operation. Taking the most conservative numbers (the maximum power output we observed of 2.1 μW , the longest experiment duration used of 2539 min, and the smallest volume of electrolyte sample used of approximately 0.5 mL), and using the density and specific heat capacity of MPN (0.937 g mL^{-1} and 2.12 $\text{J g}^{-1}\text{K}^{-1}$ respectively), we find that this power level and duration would cause an increase in temperature in the electrolyte on the order of 0.3 K. This is around two orders of magnitude smaller than our applied temperature differential. Moreover, the power being supplied to the electrolyte to maintain the temperature distribution can be (very roughly) quantified by the electric power supplied to the heating element for the hot gas supply, which is on the order of 31 W.

Finally, one can find some experimental evidence in the literature for temperature distributions in thermocells remaining constant over long durations of operation. See for example Figure S19 in reference [11], where after a temperature gradient is established it remains stable over a period of roughly 10 h. We acknowledge Reviewer 3 for alerting us to this work (see below).

b. What is the temperature range of the results in figure S7a – this should be isothermal at 25°C but it appears there is some spatial variation in temperature. Is this variation physically occurring or is it due to measurement?

As pointed out by the reviewer, some apparent variation in the electrolyte temperature (that should nominally be uniform) is visible in Figure S7a (now Figure S16a). The variations seem to occur mostly at the top and bottom of the electrolyte, and this leads us to believe that these variations are a result of experimental artifacts potentially arising from the magnetic susceptibility differences between the electrolyte and the surrounding media (air at the top and PEEK at the bottom). For this reason we have been cautious throughout our study in interpreting features in these regions. We have now added a note to this figure caption to clarify this point. Future cell designs could potentially incorporate susceptibility-matched materials to compensate for this issue, while more advanced MRI modalities that are less sensitive to these effects (such as SPRITE) could also be used.

c. What is the accuracy of the temperature results in any given pixel? What is the accuracy of the heating and cooling streams in terms of keeping an isothermal boundary at the electrode surface?

Based on our repeat measurements carried out on each cell orientation, the temperatures we observe are reproducible to within a maximum variation of ± 2.5 °C for each pixel. We have now added this to the caption of Figure 3.

The accuracy of the heating and cooling gas temperatures has not been a major concern of ours since the temperature of the electrolyte is measured directly during the experiment. However, the temperature *stability* is very important as we require the temperature map to remain constant over the duration of the imaging experiments. We have measured this stability using in situ thermocouples placed within the cell (behind each electrode), and the gas streams were found to remain constant to within ± 0.5 °C over multiple hours.

d. Can the authors explain more why the desired/set temperature gradient was not measured in the results (i.e. -20oC in exp vs. -30oC desired)?

The target temperatures were measured in benchtop tests with thermocouples situated close to the outside surfaces of the electrodes. The observed temperature on the inner side (i.e., the layer of electrolyte next to the electrode) is different due to the thermal gradient across the electrode itself, such that the inner surface of the (cold) electrode is warmer than the outer surface. We have now edited this section to clarify this point. We have also now incorporated other temperature maps that were acquired from the liquid electrolyte as a function of different target electrode temperatures into the Supplementary Information (Figure S7).

e. Have the authors considered running ex-situ experiments with embedded thermocouples to verify the MRI calibration procedure (or considered other methods of verification)?

This was already done and is now clarified in the manuscript along with additional temperature map images in the Supplementary Information, please see our responses to the previous two points.

f. In general, the temperature maps for vertical and cold-above-hot show a 1D profile in the temperature distribution while the hot-above-cold results show a stronger 2D variation in temperature (i.e. -20oC at the bottom boundary but also 20oC along the electrolyte container walls). Why is this the case?

We thank the reviewer for this observation. We suspect that this is due to the different hot and cold gas flow pathways around the interior of the cell for the hot-above-cold and cold-above-hot set ups. This was unfortunately unavoidable due to the restrictions in gas inlet placement as we mention in the discussion of the cell designs. We now mention this in this discussion of Figure 7.

4. The overall measurement accuracy of the method should be commented on for the concentrations and temperature. Can the authors further comment on expected repeatability of the method?

We have addressed the accuracy of the reported temperatures already in our responses above, and we now mention these in the discussion of the cell designs. With regards to the accuracy in the Co^{2+} concentration maps, these are generated from the T_1 relaxation times measured from each pixel, which are measured to a fairly high accuracy (<1% error). The largest source of error in the concentrations will therefore stem from the conversion of these measured T_1 times using the fit of the 3D data plot in Figure 2f. These calibration T_1 measurements should also be quite accurate (<1% error), so the largest source of error therefore comes from the temperature map. The 3D plot in Figure 2f

translates our estimated uncertainty in temperature for each pixel of ± 2.5 °C into an uncertainty in Co^{2+} concentration of approximately ± 0.008 M, and this is in good agreement with our observed variation (per pixel) over repeat measurements. We have now added this to the caption of Figure 4.

5. What were the number of averages used for the different sequences? 1H relaxation maps took 27 minutes, yet TR was varied from 100-6000ms, which will change the duration of the scan. The 19F images were acquired with a TR of 20s and 'additional signal averaging was employed resulting in an acquisition time of 43 minutes.' At TR=20s, this would allow for only 2 averages, less than that implied by the 27 min scan time of the 1H images.

We have now edited the methods section to clarify this. The ^1H T_1 relaxation maps were reconstructed from 15 T_1 -weighted images, each of which were obtained with different recycle delays (TR). The total time to run all 15 T_1 -weighted images was 27 min. Hence, 27 min per T_1 map. For the ^{19}F spin density imaging, only a single TR of 20 s was used, with 2 signal averages as the reviewer correctly states.

6. Why is only 250 minutes considered for Figure 5 while the results in Figure 4 imply transient behavior at that resistance value over that timescale?

We agree that the system represented in Figure 5 has likely not reached steady-state, and indeed we mention this in the discussion. We selected the times (and corresponding changes in load resistance) for Figure 5 to enable us to image the resulting changes in the concentration profiles, purely in order to demonstrate the efficacy of this imaging technique. Our next step will be to more judiciously choose the cell conditions (temperature differential, applied load(s), measurement times) in order to more comprehensively study the concentration changes and correlate them with the current/power output and/or modelling data. This will constitute a lengthy project and will be carried out as part of our future work, but is beyond the scope of this initial communication, which is focused on demonstrating the viability of the MRI technique.

7. The authors note thermal convection in the liquid cell in Figure 3a. Was this measured by obtaining the velocity profile, inferred from the temperature results, or consistent with previous modeling?

We have not yet carried out velocity mapping experiments, though this is an obvious next step to directly image convection and is work that we plan to do in the future. Convection is *inferred* from Figure 3a based on the spatial variation in temperature, which shows a more uniform temperature across the bulk of the electrolyte (consistent with rapid mixing due to convection) and also the observation of the cooler region in the middle (consistent with modelling predictions). We have now rearranged this part of the discussion to make it clear that the convection is inferred from the image based on these reasons.

8. On line 195, it is noted that convection is prevented in the gel electrolyte. However, the authors note small scale thermal convection is possible on line 291. This difference should be clarified.

We believe that some small-scale or slow convection may still occur within the pores of the gel electrolyte. To avoid a contradiction, we have now rephrased the first line that the reviewer refers to as follows:

"In the gel electrolyte under the same conditions (Figure 3b), the effects of convection on the temperature distribution are absent as a result of the polymer matrix and a more linear temperature gradient is observed across the cell."

Reviewer #3 (Remarks to the Author):

This is a very interesting system and has produced some nice measurements. My main criticism is it's over-engineered, and I find it highly unlikely anyone else in the world would adopt this form of measurement!

We respectfully disagree. The instrumentation used for this work, although certainly not commonplace, is standard commercially-available hardware that can be found in any number of MRI research laboratories and also in specialised NMR facilities set up for materials characterisation, such as ours. We are far from the only research group applying operando MRI methods to energy storage devices, and a number of prominent research groups in Europe and North America have this capability (see our various references to other studies [28-39]). Thermocell researchers wishing to use this approach should have little trouble finding willing collaborators with the necessary hardware and expertise.

It's also unfortunate that this particular form of measurement cannot be extended to the vast majority of electrolytes actually employed in thermocells. This does effect the impact of this work to a degree.

This is incorrect, and we are unsure how the reviewer came to this conclusion. We refer to our response to Reviewer 1's fourth point above, and we hope that our addition to the concluding paragraph of the manuscript clear up this misconception:

“The ^1H relaxation-based MRI approach we present in this work should be applicable to any electrolyte system containing a protonated solvent molecule (e.g., water, or organic solvents) and a redox couple where one of the redox states is paramagnetic to provide enhanced relaxation contrast. Spin density imaging, also demonstrated here using ^{19}F , may be applicable to non-protonated systems, redox couples with no paramagnetic state, or solid or mixed-phase electrolytes provided they contain magnetic nuclei with sufficient sensitivity and suitable relaxation times.”

Fundamentally, it's a nice idea, nice setup and has produced some nice results. I feel it merits publication, but have the below comments:

(1) The manuscript undersells the power of IR cameras (ref. 25). And it also doesn't mention other work involving detailed characterisation using IR cameras, such as reference 11. This makes the manuscript a bit misleading; the authors should expand upon prior work (such as ref 11) and instead point out what this technique can do that the IR camera cannot (e.g. Soret). Very clearly external IR imaging can strongly reflect the interior of the cell.

We thank the reviewer for raising our awareness of the thermal images associated with reference 11, which we had previously overlooked due to the images being contained in the supporting information of that paper. We now include mention of this work in our introduction section and in fact these results appear to back up our assertion (discussed above in response to Reviewer 2) that the spatial distribution in temperature in an operating cell remains constant during device operation.

It was not our intention to downplay the role that IR imaging can play, and IR will continue to be a useful and important tool in thermocell characterisation. However, we do believe that our method is superior to IR imaging in several specific ways (primarily its ability to resolve internal spatial differences in temperature and, crucially, concentrations), and we believe that these advantages are made sufficiently clear in our manuscript.

(2) The same extends to modelling; I believe only 2 modelling studies were mentioned, but ignores modelling such as Ref 11. A very quick search also turned up Wu et al. (2017, ELECTROCHIMICA ACTA,

2017, 225, 482-492, <https://doi.org/10.1016/j.electacta.2016.12.152>) and in the ESI of Im et al (NANO RESEARCH, 2014, 7, 443–452, <https://link.springer.com/article/10.1007/s12274-014-0410-6>); I am sure there are others.

We have now incorporated the suggested references. Our manuscript was not intended to provide an exhaustive review of previous modelling studies of thermocells, of which there are many as the reviewer points out. Rather, we selected two modelling-centric papers for discussion to highlight the kinds of spatial variations in various quantities that modelling can provide, and to which our imaging results can potentially be directly compared.

(3) The thermocell measurements conditions are poorly described; I cannot see any relevant data from the actual measurements, there's no description in the Methods, and the manuscript only contains a cryptic "Subsequently, an external load resistance is applied and the cell begins to operate, with both the current and potential continuously measured via the potentiostat." statement.

We assume that the reviewer refers to the current and voltage measurements. We did include the description of the basic potentiostat set up in our Methods section but we agree that this warrants more detail and we have now expanded this into its own separate section (Electrochemical measurements).

Throughout this work we have used a rather arbitrary series of load resistances applied at different times and these were selected based on the timings of the imaging experiments to best illustrate the changes visible in said images. In effect, the imaging protocol was prioritised and the choice and timings of the various load resistances were considered secondary. The reason for this was our focus on demonstrating the viability of the imaging technique itself rather than carrying out a systematic study of the performance of these thermocells. In the figures, we provide information on the load resistances applied and their timings, as well as the powers drawn from the cells during the acquisition of each image (average values, as the images were generally not acquired under steady-state conditions). However, in light of this comment and Reviewer 1's second point, we now also provide plots of the raw data (i.e., the thermocell voltages and currents) for all images in the Supplementary Information (Figures S11 to S15).

(4) Since potential was measured, the Nernst equation can be used to model the predicted ratio's of Co(II/III) at the electrode surfaces (cf. Buckingham et al., SUS ENERGY FUELS, 2020, 4, 3388-3399, <https://pubs.rsc.org/en/content/articlelanding/2020/se/d0se00440e#!divAbstract>). Do the values measured and plotted in Fig 5(c) match with the calculated values? Presumably any difference is due to either the effect of ratio upon the seebeck coefficient, and/or Soret.

As noted by the authors of this reference, the Nernst equation is unfortunately not applicable when a current is being drawn due to the resulting overpotential, which is the case for the data in Figure 5. Indeed, as we note in the discussion, the concentration profiles in Figure 5 are likely not even at steady state, making comparison of this data with theory very difficult. We are confident that an extensive future study with more carefully selected cell operating conditions (beyond the scope of this initial communication) will enable quantitative comparisons of experimental concentration maps with theory (and modelling results).

(5) Various areas of the R&D are not compared vs prior literature and are thus treated as novel, e.g. on page 8 it's stated that gelling the electrolyte switched off convection (something well understood and previously reported; the novelty here is imaging in), on page 9 the 'sinking effect' is noted (that's been mentioned before in a lot of the earlier fundamental work, e.g. by Ikeshoji [1987, Bull. Chem. Soc. Jpn., 60, 1505-1514]) and the hot-over-cold / cold-over-hot is introduced without any context. In

particular, the average reader will not understand how this orientation study is actually a 'failure' of the experiment to match expectations because genuinely gelled electrolytes should not have been effected by orientation - and either (a) this technique is visualising things which are hard to conceive and observe in the macro state, or (b) the gel melted!

Regarding the gel electrolyte preventing convection, we have now edited this part in light of previous reviewer comments. We do not believe that this section presents the finding as a novel result. Indeed, gel matrices preventing bulk convection is common knowledge and by itself is an unremarkable result as the reviewer points out. We believe our sentence "These images provide a stark illustration of the role that convection can play in determining the temperature distribution within a thermocell, and the ability of gel electrolytes to maintain a large temperature differential by eliminating convective flow" sufficiently highlights that it is the *imaging* of the effect on the temperature distribution that is novel, not the phenomenon itself.

Regarding the "sinking effect", we have now removed this phrase to make it clear that this is not a new phenomenon. It now reads "...gradually increasing the local electrolyte density and causing it to sink." Again, this is an unremarkable result by itself and dense regions of a liquid sinking due to gravity (convection) is too fundamental to even warrant a reference. Our focus here is on the ability of the imaging method to observe thermophoresis and other effects such as this.

To provide context for the hot-over-cold and cold-over-hot configurations, we have added the following sentence to clarify that the effects of these different orientations have been previously studied:

"The effects of such orientations on thermocell performance, and particularly the advantage of the cold-above-hot set up in promoting mass transport via thermal convection, have been previously established [21,22]."

We are confident that the system studied was gelled correctly, and that the gel did not melt during the experiment (this was verified by heating the gel on the bench). As discussed in our response to Reviewer 2's point 3f, these two cell configurations did show different temperature gradient variations which may also partially account for the different cell performances. We now mention this as a potential alternative explanation, though we believe that some small scale convection effects (whether due to temperature or concentration differences) may still occur within the pores of the gel and such effects would potentially be sensitive to the cell orientation as we have noted.

We have therefore responded to all of the points raised by the three reviewers. In light of their helpful and constructive comments, we have made a significant number of changes and additions to our manuscript and its supplementary information that we believe improve its quality and clarity. We again thank the reviewers for their careful reading and insightful feedback.

Additionally, during the process of revising the manuscript, one of our authors (Prof Maria Forsyth) has requested to be changed to an acknowledgment, as while she has been involved in the supervision of the work she has decided that her scholarly contribution to this manuscript is not sufficient to justify authorship. The other three authors accept this decision, and we have therefore updated the manuscript's author list accordingly.

REVIEWERS' COMMENTS

Reviewer #1 (Remarks to the Author):

The authors have addressed or answered my comments from the previous revision. The work is of great significance in the field of thermogalvanic and ionic thermoelectric cells. I recommend the acceptance of this manuscript.

Reviewer #2 (Remarks to the Author):

The authors addressed all of the original review comments in this revision. The authors are thanked for clarifications in the manuscript and for the additional supplemental data.

13th October 2021

Response to reviewer comments for manuscript NCOMMS-21-21865A

Reviewer #1 (Remarks to the Author):

The authors have addressed or answered my comments from the previous revision. The work is of great significance in the field of thermogalvanic and ionic thermoelectric cells. I recommend the acceptance of this manuscript.

Reviewer #2 (Remarks to the Author):

The authors addressed all of the original review comments in this revision. The authors are thanked for clarifications in the manuscript and for the additional supplemental data.

We thank the reviewers for their careful consideration and helpful feedback which helped us to improve our manuscript.